# Altered Expression of BCRP Impacts Fetal Accumulation of Rosuvastatin in a Rat Model of Preeclampsia

**DOI:** 10.3390/pharmaceutics16070884

**Published:** 2024-06-30

**Authors:** Wanying Dai, Micheline Piquette-Miller

**Affiliations:** Leslie Dan Faculty of Pharmacy, University of Toronto, Toronto, ON M5S 3M2, Canada; wanying.dai@mail.utoronto.ca

**Keywords:** breast cancer resistance protein, preeclampsia, pharmacokinetic, pregnancy, transporters

## Abstract

Expression of the breast cancer resistance protein (BCRP/ABCG2) transporter is downregulated in placentas from women with preeclampsia (PE) and in an immunological rat model of PE. While many drugs are substrates of this important efflux transporter, the impact of PE associated BCRP downregulation on maternal and fetal drug exposure has not been investigated. Using the PE rat model, we performed a pharmacokinetic study with rosuvastatin (RSV), a BCRP substrate, to investigate this impact. PE was induced in rats during gestational days (GD) 13 to 16 with daily low-dose endotoxin. On GD18, RSV (3 mg/kg) was administrated intravenously, and rats were sacrificed at time intervals between 0.5 and 6 h. As compared to controls, placental expression of Bcrp and Oatp2b1 significantly decreased in PE rats. A corresponding increase in RSV levels was seen in fetal tissues and amniotic fluid of the PE group (*p* < 0.05), while maternal plasma concentrations remained unchanged from the controls. An increase in Bcrp expression and decreased RSV concentration were seen in the livers of PE dams. This suggests that PE-mediated transporter dysregulation leads to significant changes in the maternal and fetal RSV disposition. Overall, our findings demonstrate that altered placental expression of transporters in PE can increase fetal accumulation of their substrates.

## 1. Introduction

The use of medications during pregnancy is progressively increasing, with more than half of pregnant women reporting the use of either prescription or over-the-counter drugs during gestation [1]. Due to physiological changes that occur during pregnancy, the pharmacokinetics of commonly used medications can be altered. Some of these modifications include increases in cardiac output, total body water, fat compartment, renal blood flow, and alterations in plasma albumin and CYP450 activity [2]. Pharmacokinetics can be further altered by pathophysiologic states. For instance, preeclampsia (PE) patients have significantly lower levels of albumin, which can lead to decreased plasma protein binding and altered distribution of drugs [3]. As PE affects 3–8% of pregnant women worldwide and is the leading cause of maternal and fetal morbidity [4], it is essential to have a comprehensive understanding of the pharmacokinetic alterations caused by PE and the possible adverse effects on the developing fetus. However, there is a lack of knowledge regarding the impact of maternal diseases, such as PE, on drug disposition. Insufficient understanding of the factors that may alter maternal and fetal drug exposure can pose significant risks to both mothers and fetuses.

Many drugs can cross the placental barrier, hence increasing the risk of fetal exposure and potential teratogenic outcomes. While transporters have been shown to play an important role in fetal drug exposure, we have found that maternal diseases such as HIV and chorioamnionitis are associated with altered expression of transporters within the placenta [5,6]. Of note, expression of the breast cancer resistance protein (BCRP/ABCG2) was significantly decreased by 60% in placentas obtained from women diagnosed with PE [7]. The apical expression of BCRP in placental syncytiotrophoblast cells limits fetal exposure to drugs and xenobiotics by effluxing them back into the maternal circulation. Indeed, studies have shown that Bcrp knockout mice have 5-fold increased fetal exposure to the Bcrp substrate nitrofurantoin compared to wild-type mice, thus demonstrating the protective role of BCRP in limiting fetal exposure to xenobiotics [8]. A decrease in placental BCRP expression could therefore result in the accumulation of BCRP substrates on the fetal side, leading to potential teratogenic effects. In addition to BCRP, P-glycoprotein (ABCB1), and organic anion transporting polypeptide 2B1 (OATP2B1/SLCO2B1) are also highly expressed in human placentas and play a crucial protective function in fetal development [9]. These transporters have also been observed to be dysregulated in the placentas of patients with PE, although to a lesser extent [7,10,11].

While PE patients generally require drug therapy to manage their disease, many therapeutic agents are substrates of BCRP. For example, nifedipine, commonly used in PE patients to control hypertension, is a substrate for BCRP [12]. As a pronounced decrease in BCRP expression was seen in PE women, our aim was to examine the impact of PE on maternal and fetal exposure to drug substrates. The ability to perform pharmacokinetic studies in pregnant women is limited due to ethical considerations; therefore, in vivo studies in preclinical animal models are needed in order to help us understand how diseases such as PE affect maternal and fetal drug exposure. We recently characterized an immunological rodent model of PE that demonstrates similar downregulation of placental Bcrp to that seen in human PE, along with comparable phenotypic changes such as proteinuria and immune activation [13]. This frequently used rodent model involves the administration of lipopolysaccharide to induce an immune response that results in impaired spiral artery remodeling, elevated blood pressure, increased levels of proinflammatory cytokines and renal involvement, thus sharing many features with early-onset PE.

Rosuvastatin (RSV) is a BCRP substrate recommended by the FDA for in vivo studies as it undergoes minimal metabolism (<10%) and has limited permeability via passive diffusion [14,15]. RSV belongs to a group of HMG-CoA reductase inhibitors, also known as statins, which are commonly used to lower cholesterol levels and prevent cardiovascular disease. Besides BCRP, RSV is also transported by OATP2B1 in the intestine and by the liver-specific OATP1B1, and OATP1B3 transporters [16]. For the quantification of RSV concentrations in placental tissues, we employed liquid chromatography–tandem mass spectrometry (LC-MS/MS). This method offers high sensitivity and specificity, making it suitable for detecting low levels of drugs in fetal tissues and amniotic fluid. Using the immunological model of PE, we conducted a comprehensive pharmacokinetic and biodistribution study of RSV in PE and control dams after IV administration. This information could ultimately assist in the development of therapeutic guidelines for the management of maternal diseases.

## 2. Materials and Methods

### 2.1. Animals

All animal studies were approved by the Office of Research Ethics at the University of Toronto and conducted in accordance with the guidelines of the Canadian Council on Animal Care. Timed pregnant Sprague Dawley rats (Charles River Laboratories, Sherbrooke, QC, Canada) were received and acclimatized on GD11 and housed in a 12-h light–dark cycle with free access to water and food. The model of PE was previously established and further characterized in our laboratory [13]. Briefly, pregnant rats were injected intraperitoneally with 0.01 mg/kg of endotoxin (lipopolysaccharide, Escherichia coli serotype 0111:B4; Sigma-Aldrich, Oakville, ON, Canada) on GD13, followed by daily 0.04 mg/kg doses on GD14, GD15 and GD16. The controls received saline. Urine was collected on GD13, 16, and 18, prior to any other procedures.

On GD18, RSV (3 mg/kg) was administrated intravenously via tail veins in control (CT) and PE rats. After 0.5 h, 2.5 h, 4 h, and 6 h, rats were anesthetized and sacrificed. Time points were chosen based on a pilot pharmacokinetic study that examined plasma and tissue concentrations of RSV in healthy pregnant rats from 0.25 h to 8 h. The pilot study results found that the concentrations of RSV in maternal plasma were below the detection limit (1 ng/mL) at 8 h, while levels in fetal tissues and amniotic fluid were below the detection limit at 0.25, 6, and 8 h. Thus, based on these time-concentration data, we chose to examine RSV distribution at 0.5, 2.5, 4, and 6 h. Cardiac puncture was used to collect maternal blood, followed by centrifugation at 1500× *g* and 4 °C to isolate plasma. Maternal plasma, livers, kidneys, placentas, fetuses, and amniotic fluid were collected, snap-frozen in liquid nitrogen, and stored at −80 °C for future analysis. Each time point included 4 PE and 4 CT animals, except for the 6 h time point, which included 4 PE and 3 CT, for a total of 16 PE and 15 CT animals.

### 2.2. Plasma and Urine Analysis

In order to confirm the development of disease in the PE group, we measured the maternal plasma protein expression of IL-6 and urinary total protein levels. Protein expression of IL-6 in maternal plasma was quantified using the rat IL-6 Quantikine ELISA Kit (#R6000B, R&D Systems Inc., Toronto, ON, Canada) following the manufacturer’s protocols. Total protein concentrations were measured in collected urine using a protein assay (Bio-Rad Laboratories, Mississauga, ON, Canada) and were normalized against total urinary creatinine concentrations using a Creatinine Colorimetric Assay Kit (Cayman Chemical Company, Ann Arbor, MI, USA).

### 2.3. Quantitative Real-Time Polymerase Chain Reaction (qPCR) Analysis

Previously reported PE-mediated changes in transporter mRNA expression were confirmed via qPCR, as previously described [13]. Briefly, total RNA was isolated from tissues using TRIzol reagent (Invitrogen) according to the manufacturer’s instructions. RNA concentration and purity were measured by a NanoDrop 1000 spectrometer (Thermo Fisher Scientific, Mississauga, ON, Canada). Isolated RNA was treated with DNase I, reverse transcribed to cDNA using a High-Capacity cDNA RT Kit and quantified using the Power SYBR Green detection system (Applied Biosystems, Thermo Fisher Scientific, Mississauga, ON, Canada). Primer sequences have been previously published [13].

### 2.4. Protein Binding Assay

Protein binding of RSV was determined in plasma samples obtained from PE and CT dams at 0.5 h after administration of RSV (*n* = 4/group). A previously described ultrafiltration (UF) method was used, with slight modifications [17]. UF units (Ultrafree-MC regenerated cellulose membrane, MWCO 10K) were purchased from Millipore (Bedford, MA, USA). Briefly, UF units were pretreated with 25 μL of 5% Tween 80 for 5 min, centrifuged for 10 min at 3000× *g* and washed with 200 μL PBS. Plasma samples were then added to the pretreated filter membrane and incubated for 1 h at room temperature. Plasma samples were centrifuged at 3000× *g* for 20 min to collect 200 μL of filtrate. The internal standard was added to filtrate samples and subsequently analyzed by LC-MS-MS as described below.

### 2.5. Sample Preparation

The method of sample preparation has been previously described [18]. Briefly, 100 μL of internal standard solution (1000 ng/mL carbamazepine in methanol) was added to 100 μL PE and control plasma samples and vortexed. Subsequently, 4 mL of ethyl acetate was added for extraction, followed by vortexing at 2000 rpm for 5 min. The samples were centrifuged for 5 min at 8000× *g*, and the upper organic phase was transferred to another glass vial, evaporated to dryness under a nitrogen evaporator and the residue reconstituted in 200 μL of 0.5% acetic acid in water/methanol (50:50, *v*/*v*). Samples were transferred to glass 250 µL autosampler vial inserts and 40 µL of the samples was injected into the chromatographic system.

For tissue samples, approximately 200 to 300 mg of fetus, placenta, liver or kidney was homogenized with 1.1 mL of 1 M acetic acid/methanol (50:50, *v*/*v*). After homogenization, 100 μL of 0.5% acetic acid, 600 μL of 0.5% tetrabutyl ammonium hydroxide, and 100 μL of internal standard solution were added and samples were extracted and prepared similar to that described for plasma. Standard curves for RSV concentrations were prepared using plasma and tissues from non-treated pregnant rats.

### 2.6. Measurement of RSV Concentrations Using LC-MS-MS

The concentrations of RSV in rat plasma, placenta, fetal tissues, amniotic fluid, and maternal tissues were measured using LC-MS-MS as described by Lan et al. [18]. A mobile phase for HPLC consisting of methanol and 2% formic acid in water (80:20, *v*/*v*) was delivered at a flow rate of 0.55 mL/min. A ZORBAX Eclipse XDB-C18 column 5 μm (4.6 × 150 mm) was maintained at a temperature of 20 °C. The injection volume was 40 μL. The mass spectrometer (Agilent 1100 Series LC/MSD, Agilent Technologies, Mississauga, ON, Canada) was operated in the positive ion mode, and the vaporizer temperature set at 450 °C. The specific transition *m*/*z* 482→258 amu was employed for analyzing RSV, with a dwell time of 200 ms. For the internal standard, carbamazepine, the mass transition *m*/*z* 237→194 was used with the same dwell time. The ionspray voltage was set to 5000 V, the decluster potential was set to 120 V and the collision energy to 25 V. The collision cell exit potential was 5 V. The nebulizer gas pressure and curtain gas pressure were set to 8 and 7 psi, respectively, at 550 °C. The retention time was set at 3.38 min and 3.53 for RSV and internal standard, respectively. Chromatographic peaks were evaluated using Agilent ChemStation (E02). The lower limit of quantification of RSV was 1 ng/mL in plasma and amniotic fluid and 0.4 ng/g in tissues.

### 2.7. Data Analysis

Extrapolation of plasma concentration to time 0 and pharmacokinetic parameters were estimated by non-compartmental analysis using PKsolver (https://www.pharmpk.com/soft.html). Statistical significance between PE and CT was measured by GraphPad Prism8 using a two-tailed Student’s *t*-test for unpaired experimental values. Significance was set to *p* < 0.05. All results are presented as mean ± standard deviation (S.D.).

## 3. Results

### 3.1. Confirmation of PE and Transporter Downregulation in PE Rats

We confirmed PE-associated phenotypic disease characteristics and transporter changes in samples obtained from the dams used for our pharmacokinetic study. There was an elevation in urine protein levels from the PE rats on GD16, 3 days after beginning endotoxin injections on GD13 (Appendix A). The urinary ratio of total protein to creatinine was significantly higher in the urine of PE rats on GD16 and 18, while no difference was detected on GD13. Relative plasma concentrations of albumin were decreased by approximately 20% in the PE group (control: 100 ± 23% versus PE: 77.8 ± 14%; *p* < 0.05). In addition, significantly elevated levels of IL-6 were detected in the maternal plasma of PE rats on GD18 (Appendix A).

Transcript levels of Bcrp and Oatp2b1 were significantly decreased in the placentas of the PE dams used in the pharmacokinetic study (Figure 1A), while hepatic transcript levels of Bcrp were significantly increased (Figure 1B). Oatp1b2 was significantly decreased in livers of the PE dams (Figure 1B). Corresponding Western blot data for the hepatic transporter expression can be found in the Appendix A.

### 3.2. RSV Plasma Protein Binding

Maternal plasma samples from CT and PE rats at 0.5 h post-injection were used to assess plasma protein binding of RSV as the highest plasma concentrations were detected at this time point. The total plasma concentrations of RSV in CT rats and PE rats at 0.5 h were 640 ± 40 ng/mL and 650 ± 40 ng/mL, respectively. We observed 10.6 ± 0.6% unbound RSV in CT dams, and 11 ± 1.6% unbound RSV in PE dams. There was no significant difference between the two groups.

### 3.3. RSV Pharmacokinetics in Maternal and Fetal Compartments

Maternal plasma concentration–time profiles of RSV were not significantly different between PE and CT dams (Figure 2A). Pharmacokinetic parameters of RSV in maternal serum are summarized in Table 1. In fetal tissue, we observed RSV accumulation between 0.5 and 4 h (Figure 2B), with subsequent elimination after 4 h, which was consistent with the decreasing maternal plasma concentrations. Although the maternal plasma concentration–time profile was similar in CT rats and PE rats, the RSV concentration–time profile showed significantly higher fetal concentrations of RSV in the PE group at 4 h. While RSV concentrations in fetal tissues were only significantly different at 4 h, the overall AUC_(0–6h)_ in fetal tissue was found to be significantly higher in PE (13.8 ± 0.8 ng·h/g) than in CT (11.4 ± 0.6 ng·h/g) (*p* < 0.05), with a relative PE: CT AUC ratio of 1.21.

RSV concentration–time profiles in placenta, maternal liver, maternal kidney, and fetal amniotic fluid are shown in Table 2. Similar to that seen in fetal tissue, higher RSV concentrations were found in the amniotic fluids of the PE fetal units at 4 h along with a significantly higher AUC_(0–4h)_. The relative PE: CT AUC_(0–4h)_ ratio of RSV in amniotic fluid was 1.12. On the other hand, RSV hepatic concentrations and RSV AUC_(0–6h)_ in maternal livers were significantly decreased in the PE dams.

Concentrations of RSV in the placenta, fetus, and amniotic fluid of individual fetal units were normalized to RSV maternal plasma concentrations obtained at the same time from the same dam and the tissue:plasma concentration ratios are shown in Figure 3. PE was associated with a significantly increased accumulation of RSV in both the fetus and the amniotic fluid. As compared to CT, the ratio of RSV concentration in the fetus to plasma was approximately 70% higher in the PE group at 4h and 40% higher at 6 h (*p* < 0.05). Likewise, a 40% higher RSV concentration ratio was found in amniotic fluid from PE as compared to CT at 4 h (*p* < 0.05). RSV concentration ratios in placenta were not significantly different between PE and CT dams (Appendix A).

Tissue: plasma concentration ratios of RSV for maternal liver and kidney are shown in Figure 4. In maternal kidney, the RSV tissue concentration ratio was comparable in both CT and PE dams. However, RSV concentration ratios were 40–60% lower in livers of PE as compared to CT dams at 2.5 h and 4 h, with no significant differences seen at 0.5 h and 6 h.

## 4. Discussion

Overall, our current study demonstrated that PE-mediated changes in Bcrp expression in pregnant rats are associated with altered maternal and fetal disposition of the Bcrp substrate, RSV. This immunological rat model of PE has been previously characterized and has many phenotypic similarities to human PE [19,20], including PE-associated changes in placental transporters [7,13]. Characterization of phenotypic changes in this current study demonstrated elevated plasma concentrations of IL-6, proteinurea, and decreased placental expression of Bcrp in PE dams, which are consistent with previous reports [13,19,20].

HMG-CoA reductase inhibitors, also known as statins, are a class of drugs that are used to treat hypercholesterolemia by lowering the amount of cholesterol in the blood [21]. The HMG-CoA reductase inhibitor, RSV, is recommended by the FDA for use as a BCRP substrate drug in clinical studies [14]. In both humans and rodents, RSV has a relatively low oral bioavailability (0.05–0.2), is actively transported into the liver and excreted in bile, mostly in its original form [22]. In humans, almost 70% of its total clearance is attributed to its hepatic elimination, with only minor metabolic clearance [23]. BCRP has been shown to transport RSV both in vitro and in vivo and is primarily responsible for the secretion of RSV into bile [16,24]. Indeed, studies in knockout mice have confirmed that BCRP plays an important role in RSV disposition [21,25]. RSV has limited permeability via passive diffusion and uptake occurs primarily through the organic anion transporters including OATP2B1 and the liver-specific OATP1B1 and OATP1B3 transporters [26]. During the hepato-biliary elimination process, OATP1B1/3 is responsible for the hepatic uptake of RSV from the systemic circulation while BCRP is involved in the active secretion of RSV from hepatocytes into bile [16].

Our study demonstrated that PE was associated with 40–70% increased fetal: maternal plasma concentration ratios of RSV, as well as a 40% higher amniotic fluid:maternal concentration ratio. These findings indicate that PE-mediated changes result in an increased accumulation of RSV in fetal tissues. Likewise, the higher AUC values observed in the fetus and amniotic fluid of the PE rats indicate increased fetal drug accumulation. The plasma protein binding of RSV is estimated to be 88% and this occurs through binding to albumin [15]. Multiple studies have reported that PE is associated with reduced plasma albumin levels [3,27,28]. Previous characterization of the PE model using proteomic analysis detected both a 20% decrease in levels of albumin and 200% increase in levels of alpha-one glycoprotein in PE rats [13]. Despite these changes, we found that plasma protein binding of RSV was not significantly different between PE and control dams and therefore unlikely to be responsible for altered tissue disposition. Therefore, we believe that the increased fetal accumulation of RSV most likely occurs through changes in the placental expression of Bcrp and Oatp2b1.

Decreased placental expression of Bcrp can cause increases in the placental and fetal accumulation of its substrates. Located on the apical side of the placental snycytiotrophoblast, BCRP transports substrates from the fetal to the maternal compartment, limiting the uptake of xenobiotics into the placenta thereby decreasing fetal exposure. Previous research demonstrated that co-administration of the BCRP inhibitor GD120918 resulted in a 2-fold rise in the fetal tissue/maternal plasma concertation ratio of topotecan in pregnant mice as compared to those receiving topotecan alone [29]. Similarly, the AUC ratios of fetal tissue/maternal plasma for nitrofurantoin and glyburide were respectively found to be approximately four and two times greater in Abcg2−/− mice as compared to wide-type mice [8,30]. Since RSV is primarily transported by BCRP, the decreased expression of Bcrp in PE placenta is likely to be a major contributor to the increased fetal accumulation of RSV.

Likewise, the observed downregulation of Oatp2b1 seen in PE placenta could also contribute to altered fetal drug exposure. OATP2B1, located on the basal membrane of placenta, mediates the uptake of drugs in the fetal-to-maternal direction and therefore plays a role in removing its substrates from the fetal compartment. Thus, decreased expression would decrease fetal clearance and increase the fetal accumulation of its substrates. As RSV is a substrate of OATP2B1, the downregulation of Oatp2b1 in PE rat placentas likely contributes to the increased fetal accumulation of RSV. Interestingly, the significantly decreased placental expression of both Bcrp and Oatp2b1 did not result in significant changes to the placental accumulation of RSV. This likely stems from the combined reduction in fetal to placental transfer (due to decreased Oatp2b1) along with reduced maternal to placental transfer (due to decreased Bcrp). Indeed, it is well recognized that OATP2B1 and BCRP collaborate to facilitate the fetal-to-maternal transfer of their steroid sulfate substances across the placenta [10]. Thus, fetal-to-maternal transfer would be diminished if both transporters were compromised. It is also important to note that the placental transport of RSV could be influenced by a complex interplay of multiple transporters. While BCRP and OATP2B1 are the major RSV transporters, other transporters could compensate for the changes in their expression, thus contributing to the placental accumulation of RSV. Further research is necessary to identify and characterize these compensatory mechanisms and their role in maintaining drug levels in the placenta.

Another key finding in this study was the significantly lower hepatic concentrations and liver:plasma concentration ratio of RSV in PE dams. As RSV is a substrate for the BCRP and OATP transporters, decreased hepatic levels of RSV could stem from PE-mediated changes in hepatic expression of either Bcrp or Oatp1b2. In rats, Oatp1b2 is homologous to human OATP1B1/3 and this isoform has been shown to transport RSV [31]. Thus, the observed decrease in Oatp1b2 expression in PE could result in decreased hepatic uptake of RSV [31]. However, decreased hepatic uptake would also result in an increased plasma concentration of RSV, which was not seen. On the other hand, higher expression of hepatic Bcrp in PE dams would result in an increased secretion of RSV into bile. This would promote RSV clearance and could account for reduced liver concentrations. Hence, increased hepatic expression of BCRP in PE rats is likely responsible for the observed decreases in hepatic RSV accumulation.

In conclusion, our study provides evidence of altered maternal and fetal disposition of RSV in a well-characterized rodent model of PE; and changes could be primarily attributed to altered expression of the Bcrp and Oatp transporters. Individuals diagnosed with PE are commonly prescribed antihypertensives, anticonvulsants, or corticosteroids to manage the progression of PE or other co-existing conditions. As many clinically important drugs are substrates of BCRP or OATPs, it is essential to consider the potential for increased fetal exposure and drug–disease interactions in the therapeutic management of this disease. Further investigations are warranted to establish clinical implications for PE patients.

## Figures and Tables

**Figure 1 pharmaceutics-16-00884-f001:**
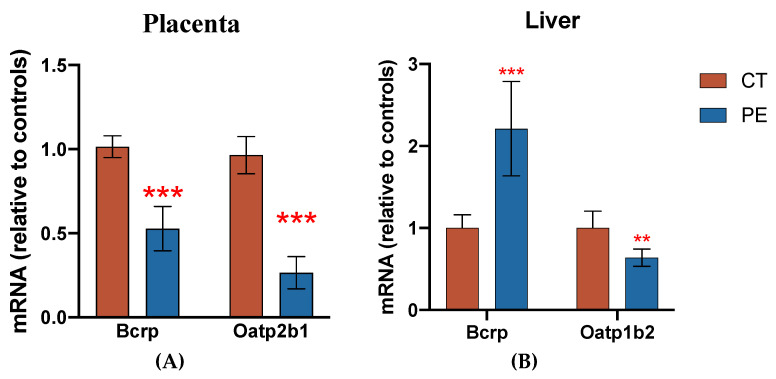
Transporter expression in CT and PE rats. Relative mRNA levels were measured in placenta (**A**) and liver (**B**) from CT and PE dams using qRT-PCR and normalized to GAPDH as described in the methods. Data are presented as mean ± S.D. and shown relative to controls (*n* = 15–16/group); ** *p* < 0.01; *** *p* < 0.001.

**Figure 2 pharmaceutics-16-00884-f002:**
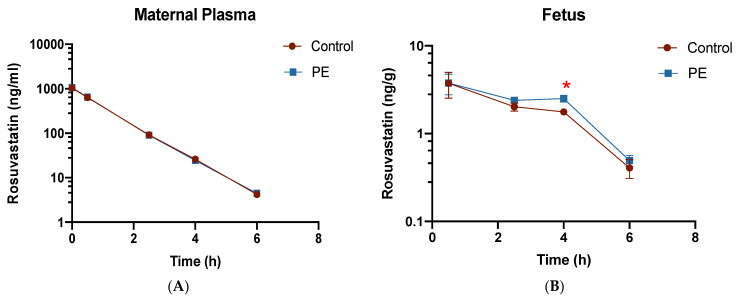
Concentration–time profiles of rosuvastatin in (**A**) maternal plasma and (**B**) fetal tissue. PE and control dams received a bolus IV dose of RSV (3 mg/kg) as described in the methods. Plasma concentrations were extrapolated to time zero using non-compartmental analysis. Data are shown as mean ± S.D; *n* = 4/group/time point except 6 h (3 CT and 4 PE); * *p* < 0.05.

**Figure 3 pharmaceutics-16-00884-f003:**
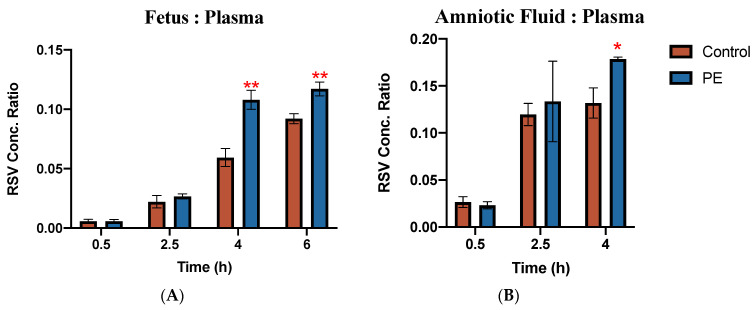
Rosuvastatin tissue: plasma concentration ratios in (**A**) fetal tissue and (**B**) amniotic fluid of control rats and PE rats. Fetal concentrations (ng/g) were normalized to maternal plasma concentrations (ng/mL) at the same time point. Data are shown as mean ± S.D.; * *p* < 0.05; ** *p* < 0.01.

**Figure 4 pharmaceutics-16-00884-f004:**
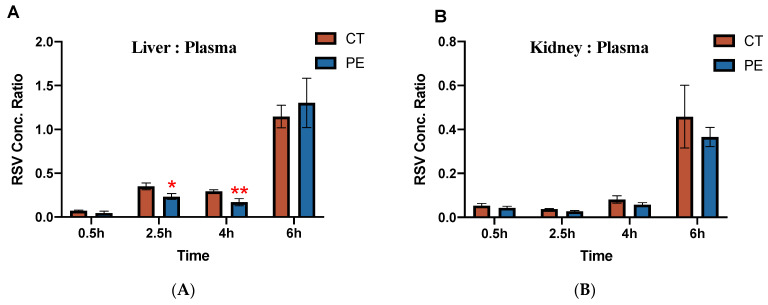
Rosuvastatin tissue: plasma concentration ratios in maternal (**A**) liver and (**B**) kidney of control rats and PE rats. Tissue concentrations (ng/g) were normalized to maternal plasma concentrations (ng/mL) at the same time point. Data are shown as mean ± S.D.; * *p* < 0.05; ** *p* < 0.01.

**Table 1 pharmaceutics-16-00884-t001:** Maternal pharmacokinetic parameters in plasma.

Group	t_1/2_ (h)	AUC_(0-6h)_ (ng·h/mL)	CL_tot_ (mL/min/kg)	V_dss_ (L/kg)
**CT**	0.8 (±0.2)	1270 (±42)	39 (±3)	1.9 (±0.1)
**PE**	0.8 (±0.2)	1278 (±71)	39 (±4)	1.9 (±0.2)
**Ratio**	1.02	1.00	0.99	0.98

Pharmacokinetic parameters in preeclampsia (PE) and control (CT) rats. Data was estimated by non-compartmental analysis using PKsolver and shown as mean (±SD). No significant differences were seen.

**Table 2 pharmaceutics-16-00884-t002:** Rosuvastatin tissue concentrations ^#^.

Time (h)	Placenta (ng/g)		Amniotic Fluid (ng/mL)		Liver (ng/g)		Kidney (ng/g)	
	Control	PE	Control	PE	Control	PE	Control	PE
**0.5**	70.8 (±9)	74.6 (±7.3)	17 (±1)	17 (±4)	46.3 (±7.4)	30.1 (±15)	33.9 (±11.5)	27.8 (±6.0)
**2.5**	20.6 (±5.3)	28.1 (±3.6)	11 (±0.6)	14 (±0.8)	32 (±2.4)	21 * (±4.2)	3.4 (±0.5)	2.5 (±0.4)
**4**	19.6 (±6.8)	15.8 (±4.1)	3 (±0.8)	4 * (±0.2)	7.6 (±0.4)	4.2 * (±1.2)	2.1 (±0.3)	1.4 (±0.2)
**6**	5.5 (±0.8)	4.5 (±2.2)	N/A	N/A	4.8 (±0.5)	5.8 (±0.9)	1.9 (±0.4)	1.6 (±0.3)
**AUC_0-6h_ (ng·h/g)**	188.4 (±8.2)	198.4 (±11.7)	48 (±1)	53 * (±2)	144.7 (±5.5)	95.8 * (±4.8)	69.0 (±10.7)	55.9 (±8.9)
**AUC Ratio**	1.05		1.12		0.66		0.81	

^#^ Data were obtained using non-compartmental analysis. Data are shown as mean (±SD); *n* = 4/group/time except 6 h (3 CT, 4 PE). * *p* < 0.05.

## Data Availability

Data are available upon request from the authors.

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
