# Peer review of "Altered Expression of BCRP Impacts Fetal Accumulation of Rosuvastatin in a Rat Model of Preeclampsia"

_pharmaceutics, 2024, doi:10.3390/pharmaceutics16070884_

Round 1

Reviewer 1 Report

Comments and Suggestions for Authors

Please, see attached file

Reviewer 2 Report

Comments and Suggestions for Authors

The paper entitled: “Pharmacokinetics and Tissue Distribution of Rosuvastatin in a Rat Model of Preeclampsia” deals with the measurements of the drug rosuvastatin (RSV) both in pregnant and fetal rat tissues and plasma. In addition the m-RNA expression of two potential statin substrate proteins, i.e.  bcrp and oatp2b1 were measured. These data are interesting since they provide interesting observations about the potential for increased fetal exposure and drug-disease interactions in the therapeutic management of the Preeclampsia disease. 

Despite presenting an interesting work, its presentation in the paper is in need of a better shape and appears often quite confounding.

Below are the main issues:

1) there is a dyscrasia between the title of the paper and its content. The title seems very focussed on  RSV and tells nothing about receptor expression profiling that is very much used in the paper content, instead.

We suggest a better coherence between the title and the content.

2)  the abstract is a mixture of an introduction and an abstract. We suggest the authors orderly summarize their work in the abstract and provide fundative information in the Introduction.

3) the introduction needs to report all relevant introductory information about the model, the drug and its interaction with biological substrates. This information appears much lately (mainly in Discussion). We suggest providing a better framework. Also analytical methods from which the validity of experimental data stems, must be introduced with their pros and cons briefly in the Intro.

4) The aim of the study and design must be clearly stated in the Abstract and Introduction and the choice of the model correctly framed, also in the context of an ethical animal suffering context, despite clear ethical committee approval has been provided. 

5) An at a glance diagram of the model and the study would greatly help understanding at a glance the study and its interest for the readers.

6) Transporter down-regulation has been provided by m-RNA expression. This is quite similar to actual protein expression but a western-blot or another kind of direct protein measurement (immuno-histology??) should be performed in order to confirm actual transporter expression.

7)In Figure 2 b the data trend is very irregular. Authors should provide a justification for that.

8) Discussion should be strongly revisited.

We suggest taking up all results in an ordered way, underscoring strengths and limitations. All statements should be carefully supported by data or, at least, citations. There are many assumptions lacking this kind of support.

10) For instance: “Dysregulation of transporters'' is a generic feat and an obvious one. They should orderlying analyze what happens in the biological system  and provide evidence for conclusions.

As it is, the paper is telling very little, apart from the fact that a biological system is perturbed by a drug and that this must be taken into account in pharmacology. But this is an obviousness and surely further to the point investigation is needed.

Round 2

Reviewer 1 Report

Comments and Suggestions for Authors

Authors adequately addressed the issues raised by the reviewer.

Reviewer 2 Report

Comments and Suggestions for Authors

The revised version of the manuscript is fine and issues we detected in the previous version have been resolved.